# Determining the Presence and Size of Shoulder Lesions in Sows Using Computer Vision

**DOI:** 10.3390/ani14010131

**Published:** 2023-12-29

**Authors:** Shubham Bery, Tami M. Brown-Brandl, Bradley T. Jones, Gary A. Rohrer, Sudhendu Raj Sharma

**Affiliations:** 1Department of Biological Systems Engineering, University of Nebraska-Lincoln, Lincoln, NE 68583, USA; sbery2@huskers.unl.edu (S.B.); raj.sharma@unl.edu (S.R.S.); 2Genetics and Breeding Research Unit, USDA-ARS U.S. Meat Animal Research Center, Clay Center, NE 68933, USA; brad.jones@usda.gov (B.T.J.); gary.rohrer@usda.gov (G.A.R.)

**Keywords:** shoulder lesions, ulcers, sows, deep learning, YOLO, U-Net

## Abstract

**Simple Summary:**

Shoulder lesions present a significant welfare concern, particularly among breeding sows. They are frequently linked to decreased mobility and loss of body condition during lactation. This research explores using RGB cameras and the potential of different computer vision methods for detecting and estimating their size. Findings indicate that these techniques hold promise in effectively identifying and quantifying lesion size. This could empower producers to proactively monitor sow welfare, facilitating timely detection and intervention for these lesions.

**Abstract:**

Shoulder sores predominantly arise in breeding sows and often result in untimely culling. Reported prevalence rates vary significantly, spanning between 5% and 50% depending upon the type of crate flooring inside a farm, the animal’s body condition, or an existing injury that causes lameness. These lesions represent not only a welfare concern but also have an economic impact due to the labor needed for treatment and medication. The objective of this study was to evaluate the use of computer vision techniques in detecting and determining the size of shoulder lesions. A Microsoft Kinect V2 camera captured the top-down depth and RGB images of sows in farrowing crates. The RGB images were collected at a resolution of 1920 × 1080. To ensure the best view of the lesions, images were selected with sows lying on their right and left sides with all legs extended. A total of 824 RGB images from 70 sows with lesions at various stages of development were identified and annotated. Three deep learning-based object detection models, YOLOv5, YOLOv8, and Faster-RCNN, pre-trained with the COCO and ImageNet datasets, were implemented to localize the lesion area. YOLOv5 was the best predictor as it was able to detect lesions with an mAP@0.5 of 0.92. To estimate the lesion area, lesion pixel segmentation was carried out on the localized region using traditional image processing techniques like Otsu’s binarization and adaptive thresholding alongside DL-based segmentation models based on U-Net architecture. In conclusion, this study demonstrates the potential of computer vision techniques in effectively detecting and assessing the size of shoulder lesions in breeding sows, providing a promising avenue for improving sow welfare and reducing economic losses.

## 1. Introduction

Shoulder lesions, amongst lactating sows, are commonly seen in the swine industry. Lesions commonly develop during the first two weeks of farrowing [1]. These are formed due to the deficiency of oxygen to the underlying shoulder tissue caused by pressure incited from the flooring. The tissues lose blood supply and die similar to human pressure ulcers [2]. The anatomy of a sow’s scapula bone has a large ridge known as the scapular spine. When the differences in the structure of the scapula in sows were analyzed, sows with a prominent scapula spine were found to be at a higher risk of lesion formation [3]. When a sow lies on its side, extra pressure is exerted on the tissue surrounding the spine making it vulnerable to ulcer formation [4]. Though the lesions are described as shoulder sores, they develop near the dorsal aspects of the spine of the scapula, not at the scapulohumeral joint [5]. The dorsal aspect of the spine is a common location of ulcer development, but lesions may develop over any bony prominence such as the tarsus or cubital joint. The severity of lesions can vary from mild lesions to bone-deep ulcers if left untreated. They are associated with poor animal welfare because of the pain and increased infection risk that can lead to euthanizing the animal. In addition to the welfare-associated problem, shoulder lesions increase production costs due to the treatment and increased culling rate [6]. Bone-deep ulcers often negatively affect the sow’s carcass value as they lead to failure of the final quality check. Therefore, treating these ulcers early to reduce welfare and economic concerns becomes highly important. Early lesion detection will result in timely veterinary treatments, such as the application of zinc oxide to the affected area. Most research regarding shoulder lesions has primarily concentrated on potential causes, associated effects, and strategies to prevent or manage ulcerations. Implementing automatic monitoring technology for shoulder lesions has the potential to alleviate the workload of farm workers and facilitate prompt treatment. In recent times, innovative animal husbandry solutions have emerged, which enable continuous and automated tracking of individual animals’ well-being through the utilization of diverse sensors and cameras [7]. However, limited research has been conducted on the monitoring of these lesions. One previous study [8] investigated the use of thermal cameras to measure the related temperature increase caused by an inflammatory response. This approach allowed for the early detection of lesions up to 7 days before visual signs appeared. These advancements not only enhance farm productivity but also enable the early detection of emerging health concerns.

While thermal imaging may be able to detect the beginning of a lesion, RGB cameras have a cost advantage. Potentially, RGB images, in combination with machine learning techniques, may be able to detect lesions shortly after they become visible and could potentially track their progression. Machine vision techniques like deep learning-based models have become increasingly effective in solving various problems. For example, convolutional neural networks (CNNs) like Resnet101, Xception, and MobileNet have already been successfully implemented to classify different postures in swine [9,10]. YOLO (You Only Look Once) has been used for the early detection of estrus behavior in cattle by modifying the spatial pyramid pooling (SPP) module in the architecture [11]. Although there has not been much work conducted in lesion detection in animals using RGB cameras, studies have been performed using deep learning models to detect and localize ulcers from the images of diabetic human feet [12]. Recently, image segmentation based on deep learning has become one of the main image segmentation methods. Different variations of U-Net architecture have been commonly used for detecting various kinds of lesions like skin [13,14], lungs [15], brain [16], etc. Another study presented a two-stage deep learning method for accurately segmenting skin lesions from dermoscopic images based on YOLO–DeepLab networks [17]. However, the images these studies used for model training had a significantly larger lesion-to-background area ratio alongside uniform illumination without much noise.

With shoulder lesions in sows, it is likely the lesion portion of the image could only be a few pixels as cameras are mounted at a height to avoid interference with day-to-day activities. In addition, for training any deep learning model from scratch, one needs to have large amounts of supervised data; for example, the pictures of sows manually annotated with bounded boxes around the lesion region to make the model understand the difference between lesion and all other objects in an image. To make these algorithms better at generalizing, focus should not only be on the dataset size but also on the data quality so that the model can extract different patterns and features from the data during the training phase.

Many applications with livestock species have the limitation of a small, annotated dataset. To tackle the limitations of a small-sized dataset, transfer learning has been used to minimize this gap. It is a machine learning technique which reuses a model trained for one task by applying it to another, related task. It uses the weights or features of a model that has been trained on much bigger datasets like MS COCO or ImageNet and feeds them into the target network. Then, modifications are performed in the initial and final layers to accommodate predictions on the custom dataset. The middle layers of any CNN learn general features of objects like specific edges, color patches, etc. and are applicable across many datasets and tasks. The target model freezes its middle layers, uses the mid-layer features from the base model, and retrains its initial and last layers to learn specific features related to the custom dataset. Jensen and Pedersen used transfer learning to localize and count pigs in slaughterhouses [18].

Therefore, the objectives of this study were to

Compare the performance of deep learning models, including two versions of YOLO (5s, 5m and 8s, 8m) and two versions of FRCNN (R50 Backbone and X-101 Backbone) in the localization of shoulder lesions in various stages of development.Compare two traditional imaging segmentation methods (Adapting thresholding and Otsu’s method thresholding) and two deep learning-based U-Net architectures (Vanilla and Attention U-net) to segment lesion pixels and estimate size.

The paper is structured as follows: Section 1, “Introduction”, outlines previous work; Section 2, “Materials and Methods”, includes details on the data collection (Section 2.1), lesion localization (Section 2.2), and segmentation processes (Section 2.3), along with size referencing techniques (Section 2.4); Section 3, “Results”, presents the findings on lesion localization (Section 3.1) and segmentation (Section 3.2) and addresses the challenges encountered (Section 3.3); Section 4 compares the findings with other studies under “Discussion” and the paper concludes in Section 5, “Conclusions”, which summarizes the key findings and insights.

## 2. Materials and Methods

### 2.1. Data Collection

The experiment was conducted at the U.S. Meat Animal Research Centre (USMARC) located outside Clay Center, NE, USA. All animal husbandry protocols were performed in compliance with federal and institutional regulations regarding proper animal care practices and were approved by the USMARC Institutional Animal Care and Use Committee (2015–2021). The facility at USMARC is a farrow to finish swine production unit. This study utilized 360 sows of a Yorkshire–Landrace cross breed. This dataset was from a study designed to test behavioural and production characteristics of sows housed in three different crate sizes [19]. The images were captured in one of the two farrowing facilities. The farrowing facility housed three farrowing rooms, with each room containing twenty farrowing crates, totaling sixty crates per farrowing cycle. The facility was well lit, with the lights being on for 12 h a day, from 5:30 a.m. until 5:30 p.m. An aluminium theatre triangle truss, 21.6 m in length, was placed above each row of crates. The bottom end of the truss was at an approximate height of 2.6 m above the ground. A time-of-flight depth sensor with an integrated digital camera (Kinect V2^™^, Microsoft, Redmond, WA, USA) was centred above each crate and mounted on the truss’s bottom at a height of 2.55 m from the floor. Sensors were enclosed within a waterproof housing to protect the cameras during pressure washing and disinfecting. Digital and depth images were collected every 5 s. Only the RGB images (resolution 1920 × 1080) were used for the experiment. The setup can be seen in Figure 1. One mini-PC with Windows 10 Home Edition (Windows 10 Home, Microsoft, Redmond, WA, USA) was connected to a single camera. A total of 20 mini-PCs were connected to an external disk station (DS1517+, Synology Inc., Bellevue, WA, USA), each having five 10 TB hard disk drives (ST10000VN0004, Seagate Technology LLC, Cupertino, CA, USA). Only images collected during the lights-on period of the day were used. The room was illuminated with 20 T-8 fluorescent bulbs.

An image capturing program was developed in MATLAB (R2017a, The Math-Works, Inc., Natick, MA, USA). This program was designed to capture and store one digital image approximately every 5 s, providing a view of lesion development across the four-week period of sow confinement in crates. Out of all the sows, 70 animals developed lesions. Within this timeframe, a total of 824 images were filtered, each depicting the progression of shoulder lesions in 70 sows. The dataset was split in the 80:20 train to test ratio.

Table 1 offers insight into the distribution of images across different weeks of lactation. Notably, data from the first week, where lesions are typically not visually evident, was also included as part of the dataset to enhance the model’s capability to handle false positives. This increase in data size aimed to enhance the model’s capability to handle false positives.

Throughout this study, the progression of lesions became apparent, with most lesions manifesting visual signs during the 1–2-week phase. These initial stages are characterized by minor wounds limited to the outer layer (epidermis) marked by reddening of the affected area. As time progressed to week 3, more severe lesions emerged, causing abrasions and reaching the lower skin layer (dermis) while growing in diameter and forming granulated tissue. By the fourth week, the most severe cases exhibited signs of deep bone lesions. Figure 2 shows the lesion progression of a severe lesion. It is important to highlight that not all the sows followed the same timeline for lesion development. Benign cases might undergo healing during later stages, often after forming scabs within the initial 2 weeks.

### 2.2. Lesion Localization

Figure 3 shows a sample image. To minimize computation for training and inference, cropping was applied because the cameras’ positions were fixed, and the animals were confined to the center of the crate. Precise identification of lesion boundaries was crucial for size determination. This was achieved using deep learning-based object detection models, which could detect lesions from sow images for localization and output bounding box coordinates of the lesion within the frame.

Two object detection architectures were tested, named YOLO (You Only Look Once) and Faster R-CNN (Region-based Convolutional Neural Network).

YOLO (You Only Look Once) approaches object detection as a regression problem. These models are renowned for their exceptional balance between speed and accuracy, making them ideal for real-time applications. This is crucial in a production setting where rapid and reliable detection of shoulder lesions is essential. The availability of various model sizes (small and medium) allowed for the tailoring of the model to the computational resources available and the optimization of either speed or accuracy. It processes entire images in one step, providing direct predictions for bounding boxes and class probabilities. This study explored two different versions of YOLO developed by Ultralytics [20] as seen in Figure 4, namely v5 and v8, with varying architecture sizes (small “s” and medium “m”).Faster-RCNN is a two-stage detector, which uses a Region Proposal Network (RPN) instead of using a selective search algorithm to output object proposals. Region-of-Interest (ROI) pooling is applied to make all proposals the same size. Then, processed proposals are passed to a fully connected layer that classifies the objects in the bounding boxes. Two different backbone models, ResNet-50 and ResNet-101, pre-trained on ImageNet classification tasks were implemented to leverage their deep residual learning framework. This is particularly advantageous for capturing complex features of shoulder lesions, which might be missed by shallower networks like YOLO.

**Figure 4 animals-14-00131-f004:**
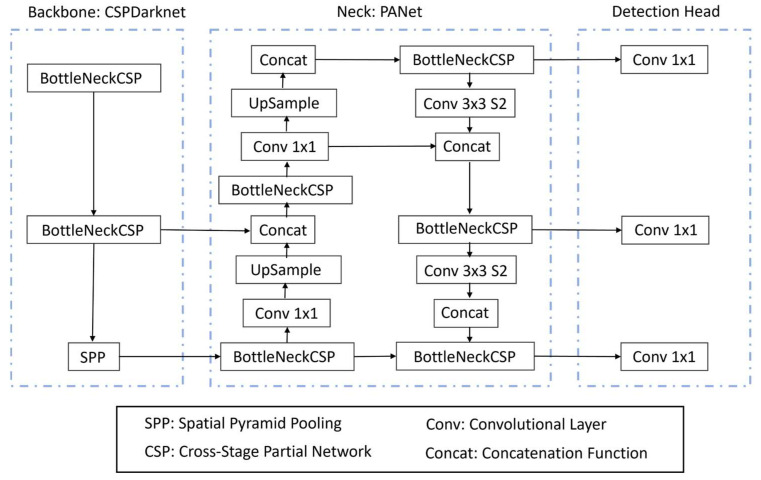
Ultralytics YOLO base architecture used for lesion localization.

Due to limitations in the dataset size, transfer learning was employed. This method fine-tuned a pre-trained detector with COCO or ImageNet weights to adapt it to a custom dataset. Both the YOLO and FRCNN models utilized the PyTorch framework (Version 1.9). Data preparation was conducted using OpenCV (Version 4.6.0), NumPy (Version 1.23), and Pandas (Version 1.5.0). Matplotlib (Version 3.6.0) was employed for data visualization. All models were trained for 300 epochs, with early stopping set to 50 consecutive iterations without performance improvement to prevent overfitting. The hyperparameters for FRCNN included a batch size of 8, a learning rate of 0.00025, and ReLU activation. For FRCNN, the default image size range of (800, 1333) pixels was used. For YOLOv5 and YOLOv8, default hyperparameters from [21] were used and the input images were resized to 1280 by 720. Object detection performance was measured by using mean average precision (mAP), which takes classification and localization into account while evaluating. The predicted bounding boxes are compared with the ground truth coordinate boxes and if the overlap between them is more than the threshold value of Intersection-over-Union (IoU), then it is considered a True Positive (TP) otherwise it gets classified as a False Positive (FP). If the model fails to detect anything when the object is there, it is a False Negative (FN). After this, precision and recall of a model are calculated. Then, the mean of all the average precision values ranging across different IoU thresholds from 0.50 to 0.95 in increments of 0.05 was calculated as shown in Equation (1); for this research, *n* = 1, as the model had to detect a single class.
(1)$$mAP=\frac{1}{n}\sum_{i=0}^{n}APi$$
where: *n* = number of classes.

### 2.3. Lesion Pixel Segmentation

Following the localization of shoulder lesions within the frames using object detection, the next crucial step involved segmenting the lesion pixels within the cropped bounding boxes. Accurate pixel-level segmentation was essential for precisely quantifying the size and extent of shoulder lesions. To obtain the segmented lesion pixels, both traditional and DL-based techniques were tested on the detected regions.

Image-processing-based binarization techniques were first used to separate out the lesion pixels from the sow’s body, as lesions were darker in color when compared to the rest of the body. Python’s OpenCV module was used to implement two automatic image thresholding techniques, namely Otsu’s method and Gaussian adaptive thresholding. Otsu’s method returns a single intensity threshold that divides pixels into two classes: foreground and background. The base value is calculated by maximizing or minimizing the intensity variance between both classes. In adaptive thresholding, different thresholds are calculated for different parts of the image for segregating all the pixels. This can handle variations in lighting due to shadows from the crate bars and multiple lesion clusters, ensuring robust pixel segmentation across the entire dataset.

Traditional image binarization might not work consistently when there is a lot of shadow noise from crate bars in the image [20] or when the variation in the lesion and the sow’s skin pixels is insufficient for it to differentiate between them, especially in earlier stages of development. Deep learning-based techniques successfully addressed these challenges where traditional image processing falls short. U-Net [22] is one such architecture that has been extensively implemented in the biomedical field for pixel-level segmentation. It is based on an encoder–decoder architecture. The vanilla U-Net alongside a variation proposed in [23] were implemented, where attention gates were introduced within the CNN architecture to make the network focus on the target object, suppressing irrelevant information within the region. Both models were initialized using ImageNet weights with frozen VGG16 backbone. ReLU was used as the activation function, with a batch size of 8, binary cross entropy as the loss function, a learning rate of 0.001, and an Adam optimizer. Both models were trained for 100 epochs.

The Dice coefficient was used for evaluating the performance of all the pixel segmentation approaches. It assesses how well a predicted binary mask aligns with a ground truth binary mask. The Dice coefficient produces a value between 0 and 1, where 0 indicates no overlap and 1 indicates a perfect match. ImageJ was used to manually annotate the lesion pixels and obtain the corresponding binary masks.
(2)$$Dice\;Coefficient=\frac{2\cdot \left|TP\right|}{2\cdot \left|TP\right|+\left|FP\right|+\left|FN\right|}$$
where

*TP* represents the number of true positive pixels, i.e., pixels that are correctly classified as lesions in both the ground truth and predicted masks.*FP* represents the number of false positive pixels, i.e., pixels that are classified as lesions in the predicted mask but not in the ground truth.*FN* represents the number of false negative pixels, i.e., pixels that are lesions in the ground truth but not in the predicted mask.

### 2.4. Size Referencing

To estimate the area covered by lesions in measurable units, a calibration was conducted using the crate’s anti-crush bar, which was positioned at the same level as the lying sow and at a similar shoulder height, serving as a reference object, as shown in Figure 5. The bar had a diameter of 25.4 mm and was 15 pixels wide. These values were utilized to calculate the area of one pixel in millimeters, which amounted to 2.87 mm^2^. By applying Equation (3), the lesion’s area was converted from pixels to mm^2^.
(3)$$Lesion\;Area\;\left({mm}^{2}\right)=\;Number\;of\;lesion\;pixels\times \;2.87\;\frac{{mm}^{2}}{pixel}$$

The entire pipeline for lesion size estimation is shown in Figure 6, starting with image cropping to lesion localization, passing the cropped bounding box to binarization models, and then performing size referencing to estimated size.

## 3. Results

### 3.1. Lesion Localization

The models were trained using a 32 GB NVIDIA Tesla V100 GPU. The main goal of this research was to identify shoulder lesions and determine their size. To achieve this, various CNN-based models for lesion localization and pixel segmentation employing image processing techniques were used. For lesion localization, Faster R-CNN (FRCNN) was trained with two different ImageNet backbones and two YOLO versions: YOLOv5 and YOLOv8, each with varying architectures. YOLO models outperformed FRCNN, as indicated in Table 2.

YOLOv5 models excelled in lesion localization, exhibiting fewer false positives, and detecting lesions in early stages of development compared to FRCNN and YOLOv8. As a result, YOLOv5m, with an mAP@0.5 of 0.92 and mAP@0.5:0.95::0.05 of 0.48, was selected for the localization task. This performance difference may be attributed, in part, to dataset size. FRCNN’s two-stage nature might require a larger dataset for optimal learning, whereas YOLO’s single-stage architecture yielded promising results even with smaller datasets, in line with [24]. After using a Python script to crop the detected lesion regions, segmentation techniques were applied to calculate the lesion area in terms of pixels.

### 3.2. Lesion Segmentation

Binarization techniques were employed to isolate lesion pixels within the cropped bounding boxes. Regarding traditional image binarization techniques, Otsu’s method exhibited better performance than Adaptive thresholding. The subpar performance of adaptive thresholding can be attributed to its sensitivity to local lighting conditions, resulting in multiple clustered segmented regions within a single cropped frame, impacted by non-uniform lighting in the crates.

During the second week of lesion progression, deep learning-based methods outperformed traditional image processing techniques. The vanilla U-Net achieved a Dice coefficient of 0.71, followed closely by its attention gates-based counterpart at 0.68, both surpassing Otsu’s method at 0.65 as shown in Table 3.

However, as lesions became more pronounced, and the distinction between foreground and background pixels became clearer, Otsu’s method outperformed all other techniques. It achieved Dice coefficients of 0.83 and 0.81 for lesions in the third and fourth weeks of development, respectively, significantly surpassing the performance of DL-based segmentation models.

The U-Net-based models showed superior performance in the early stages of the analysis. However, they encountered difficulties as the analysis progressed, as seen in Figure 7. The issue arose because these models were proficient at identifying pixels with lighter intensities as part of the lesions. Yet, as the lesions advanced, the pixels within them became darker. Despite this change, the U-Net models continued to identify the lighter skin pixels as lesions, which ultimately limited their effectiveness compared to Otsu’s method.

### 3.3. Challenges

Non-uniform crate lighting and sow movement were two big challenges faced in size estimation. Lesions were only apparent when the sow was lying on its sides. The sows’ movement impacted how the lesion area was viewed by the camera, as seen in Figure 8. The lesion pixel’s color intensity was impacted by the movement which can highly influence the segmentation process leading to variation in estimated lesion pixel count.

In addition to the sow’s movement, the bars of the crate caused occlusion, which at times made it challenging to accurately localize the lesion as it could become hidden behind these bars. Furthermore, shadows cast by these bars affected the binarization process as shown in Figure 9a. Deep learning-based methods proved more effective at segmenting pixels that included shadows, while traditional binarization techniques like Otsu’s struggled because they sometimes misclassified lesion pixels as shadows, mainly due to the limited difference in pixel intensity between the two (Figure 9b).

The binarization techniques performed poorly when the captured image had a sow being treated for lesions using a topical solution like zinc oxide. The zinc oxide ointment was the same color intensity (deep yellow) as a lesion. Algorithms mistake the applied solution for lesion pixels outputting a larger lesion area. Lesions in this study are notably relatively minor and continued to resolve not progress throughout the imaging period. This is in part due to the installation of new crates in the facility. These crates provided a large sow space for the animals. Also, all animals in this study were fourth parity or less, leading to smaller a body structure of the animals. Both factors contributed to provide appropriate comfort and improved the ability of the sows to rise, likely minimizing lesions. The floor slats were galvanized metal. The galvanization process leaves a slightly abrasive surface. This abrasiveness disappears with subsequent use but was present at the time of this study and was a contributing factor in the ethology and development of these lesions.

## 4. Discussion

This work proposed a method that advances the detection and segmentation of shoulder lesions in sows. While thermal imaging surpasses RGB in early detection, its high cost limits practicality in farm settings [8,25]. The combination of deep learning and image processing techniques offers a cost-effective alternative. Specifically, YOLOv5 models excelled in localizing lesions within frames, with the YOLOv5m model performing the best with an mAP@0.5 of 0.92 and mAP@0.5:0.95::0.05 of 0.48. It strikes a balance between speed and accuracy, making it suitable for real-time applications where both these factors are crucial, as seen in [26,27,28].

Deep learning-based segmentation models exhibited superior noise handling capabilities, effectively addressing issues like inconsistent illumination and shadows. But, U-Net models were unable to segment pixels well in the later stages, which is in contrast with [13,29,30]. It is worth noting that these studies implementing U-Net for skin lesion segmentation used microscopic images collected under uniform lighting and consistent physical conditions, unlike the conditions in a sow barn. Extending the dataset to include more images from the latter stages of lesion progression may improve the binary mask accuracy, provided there are no hardware constraints. Otsu-based binarization demonstrated strong performance in the later stages of lesion detection with Dice coefficients of 0.83 and 0.82 in the third and fourth week, respectively, where DL-based U-Net models initially performed better.

Otsu’s method had speed advantages and did not require expensive hardware implementation compared to deep learning architectures as also mentioned in [31]. To enhance its performance in the early stages, pre-binarization steps, such as increasing image contrast through histogram equalization, could further improve segmentation, which was also seen in [32]. While this may result in some information loss of lesion pixels, it can define clearer boundaries for the lesion area, ultimately improving binarization performance.

## 5. Conclusions

In summary, this study used RGB images collected on 360 lactating sows housed in typical farrowing crates. This study aimed to determine if images could be used to monitor shoulder lesion formation. This study first compared the performance of two different deep-learning models in the localization of sow shoulder lesions in various stages of development. It was determined that the YOLOv5s and YOLOv5m performed the best with a mAP@0.5 of 0.91 and 0.92, respectively. YOLOv8s and YOLOv8m models performed the second best with a mAP@0.5 of 0.84 and 0.82, respectively. Neither the FRCNN–R50 backbone nor the FRCNN–X-101 Backbone performed well with a mAP@0.5 of 0.26 and 0.56, respectively. It was hypothesized that the dataset was too small for the FRCNN models.

After identifying the lesions, this study tested traditional image processing and deep learning-based binarization techniques to estimate lesion size. The shoulder lesions observed in this study changed color as the lesions progressed. This color change resulted in changes in model performance over time. Early in lesion development, the deep-learning algorithms performed slightly better. Overall, Otsu’s method performed the best, although this model had overestimated lesion size in the early developing lesions.

This research highlights the effectiveness of using RGB images for detecting and monitoring sow shoulder lesions. However, more work needs to be completed to make this a viable Precision Livestock Farming management technique. In addition, the cost of cameras and computers may not make this approach cost-effective on its own; however, if cameras were added to the farrowing system, this technique could be one of a suite of parameters that could be monitored.

## Figures and Tables

**Figure 1 animals-14-00131-f001:**
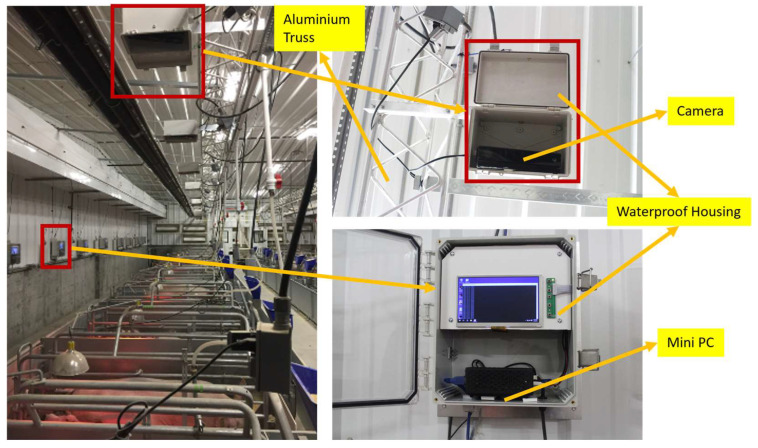
Image collection system utlizing a Kinect V2™ mounted on an aluminum truss.

**Figure 2 animals-14-00131-f002:**
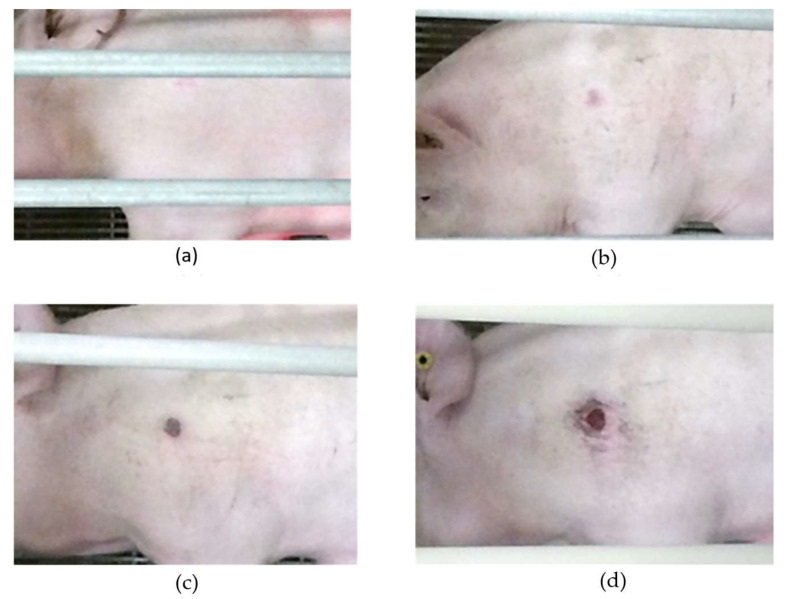
Lesion progression over the farrowing cycle. (**a**) Lesion in the first week—minimal visible signs. (**b**) Lesion in the second week—top layer affected. (**c**) Lesion in the third week—scab formed. (**d**) Lesion in the fourth week—increased lesion diameter with possible dermis damage.

**Figure 3 animals-14-00131-f003:**
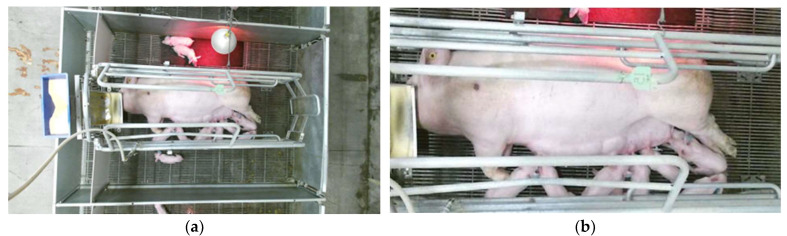
(**a**) Sample RGB images collected. (**b**) Cropped RGB image.

**Figure 5 animals-14-00131-f005:**
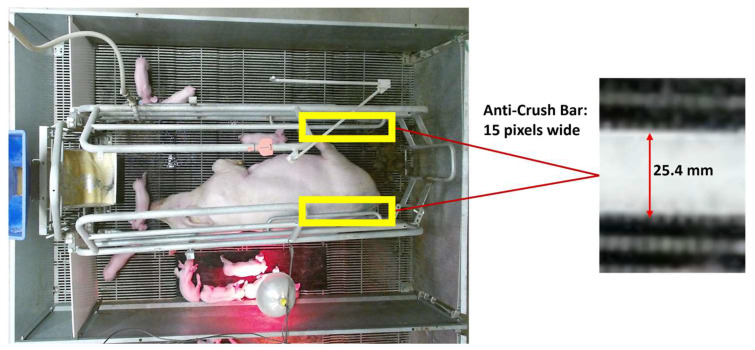
Example image to determine lesion size, the anti-crush bar was used for size referencing. The anti-crush bar was 25.4 mm in diameter equating to 15 pixels in these images.

**Figure 6 animals-14-00131-f006:**
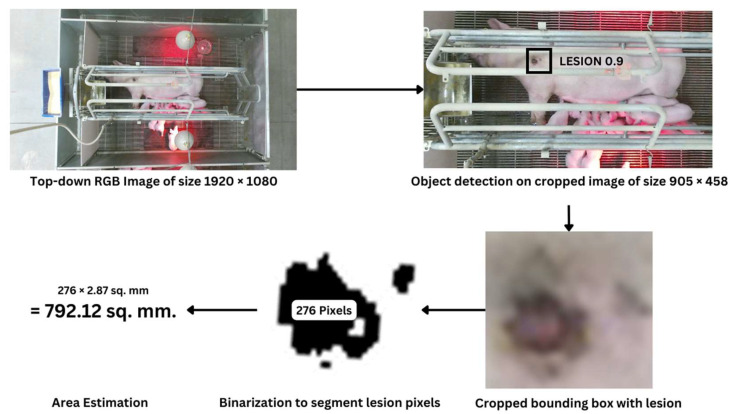
The entire pipeline showing different steps of sow shoulder lesions from image capturing to area estimation.

**Figure 7 animals-14-00131-f007:**
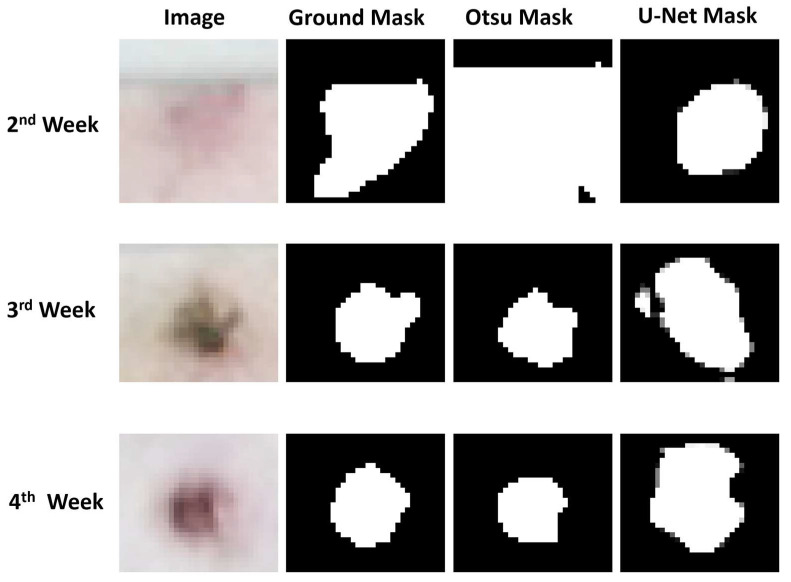
Comparing Otsu’s Method and the U-Net segmentation method with ground truth data over three weeks of lactation.

**Figure 8 animals-14-00131-f008:**
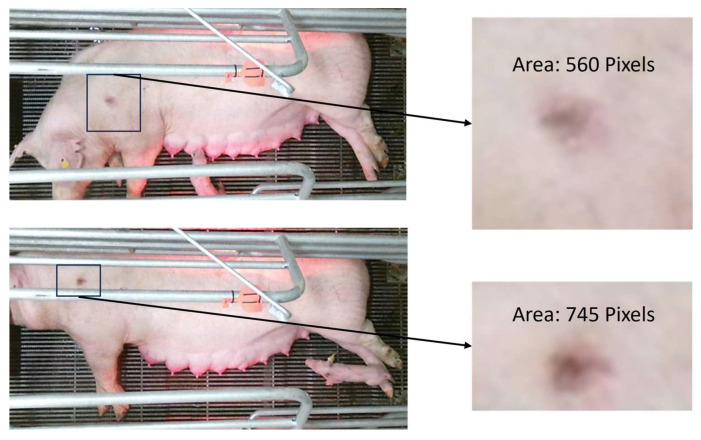
Frames captured five seconds apart where the sow changed its posture impacting the lesion appearance. Both images are sized 905 by 458 pixels.

**Figure 9 animals-14-00131-f009:**
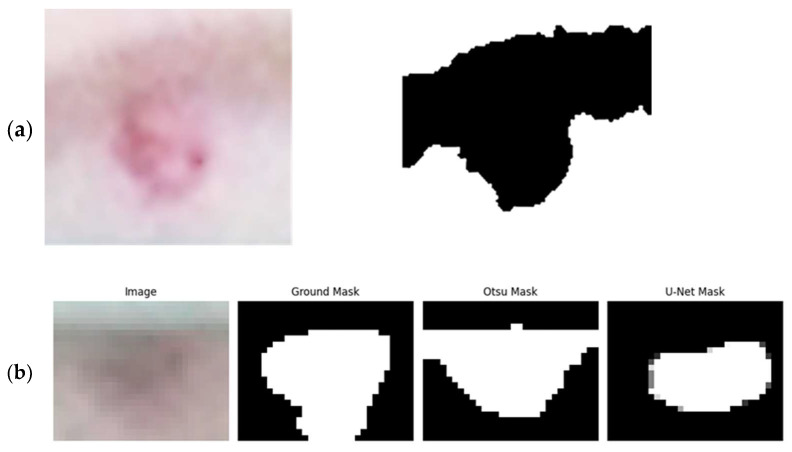
Shoulder lesion detection as impacted by (**a**) the shadow cast by the crate bar causing issues in binarization. (**b**) Comparing Otsu’s method and U-Net when the crate bar is present in the cropped region.

**Table 1 animals-14-00131-t001:** Distribution of images by week in the dataset.

Week	Number of Images
1	93
2	293
3	269
4	169

**Table 2 animals-14-00131-t002:** Comparison of performance metrics for YOLOv5, YOLOv8, and Faster R-CNN (FRCNN) object detection models. The table presents mean average precision (mAP) metrics at two different scales: mAP at a threshold of 0.5, and mAP at a threshold of 0.5:0.95, representing the average mAP calculated at IoU thresholds from 0.5 to 0.95 in steps of 0.05.

Architecture	mAP@0.5	mAP@0.5:0.95::0.05
YOLOv5s	0.91	0.46
YOLOv5m	0.92	0.48
YOLOv8s	0.84	0.31
YOLOv8m	0.81	0.35
FRCNN–R50 Backbone	0.26	0.12
FRCNN–X-101 Backbone	0.56	0.14

**Table 3 animals-14-00131-t003:** Dice coefficients for segmentation techniques across various weeks of lesion progression.

Segmentation Method	Dice Coefficient		
	2nd Week	3rd Week	4th Week
Adaptive Thresholding	0.38	0.20	0.16
Otsu’s Method	0.66	0.83	0.82
Vanilla U-Net	0.72	0.63	0.61
Attention U-Net	0.68	0.63	0.62

## Data Availability

Data are presented in this article in the form of figures and tables.

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
