# Peer review of "Determining the Presence and Size of Shoulder Lesions in Sows Using Computer Vision"

_animals, 2023, doi:10.3390/ani14010131_

Round 1

Reviewer 1 Report

Comments and Suggestions for Authors

This study aims to evaluate computer vision techniques to identify and measure the size of shoulder lesions in sows via RGB image analysis. The presented topic is interesting as a primary investigation in lesion detection using RGB images. However, the Introduction and Results section should be improved by adding more references to support the authors' statements, and more results to support the conclusions.

Specific comments

Lines 42-56: The authors should add references to the studies mentioned in the text to support these sentences.

Lines 56-57: “However, limited research has been …. “ which ones? no references available in the text. Please add references.

Lines 107-108: This sentence is unclear. Authors should improve it.

Line 111: The authors should provide more details in the introduction about the difference between “traditional” and “DL-based” binarization techniques. Also, what DL means? It would be better to specify what the acronym DL means.

Line 124: Did the authors monitor 28 crates using cameras? But in line 132, the authors state that they used “a total of 20 mini-PCs”. How can 28 farrowing crates be monitored using 20 cameras (which it is supposed that are connected to the PCs). This is confusing and the authors should clarify this point.

Line 136: In Figure 1, left part, the cameras are hard to be detected. Consider adding a magnification or highlighting the camera with an arrow.

Lines 149-162: This paragraph seems more appropriate for the discussion section.

Lines 198-200: This sentence is unclear (no verb). Consider removing “[.50:.90:.05]” as it is confusing, and it is explained in the round bracket. Moreover, it is highly recommended that the authors avoid pronouns and personal language in writing (see also line 212, 226, etc.).

 Lines 220-224: Add reference to support these sentences.

Lines 251-252: “the bar had a diameter of 25.4 cm..” but in the figure 4 the authors report 25.4 mm, which is probably the correct size. If the authors used a proportion “to calculate the area of one pixel in millimeters”, should be reported in the text. Moreover, using a proportion, the mm2 of one pixel result 2.87 and this difference could affect the final estimated size of the lesion.

Line 259: Be careful with the units of measurement in the figure caption.

Line 275: Table should be self-explanatory with in its caption. Although the meaning of "@0.5" is understandable, it would be better to explain it in the caption. Moreover, the authors state in Materials and Methods section that the mAP values were calculated ranging across different IoU thresholds, but in the table 2 they report only 0.5. Consider showing more mAP considering higher IoU also to improve the discussion of the results.

Line 350: Few results have been presented to support this claim about “determining the presence and size of lesions with accuracy and efficiency”.

Reviewer 2 Report

Comments and Suggestions for Authors

In this paper, the Authors present a solution based on computer vision and deep learning for assessing the presence and size of shoulder lesions in sows.

The document is mostly written, and it is organized as follows: Simple Summary, Abstract, a first Section containing an Introduction, a second Section on Materials and methods, a third Section on Results and Discussion, a fourth Section containing the Conclusions of the work, and finally the References used.

After a detailed review, I must say about the document that I think it might in principle result of some interest, but I also consider it could benefit from improving some aspects and highlighting its contributions regarding existing solutions. For a more detailed explanation, please, see my comments below.

1.  In Lines 19-21 the Authors state: 'A time-of-19 flight (Microsoft Kinect V2) camera captured the top-down depth and RGB images of sows in farrowing crates.' I could find no other mention in the document about such top-down depth captured. Has it been used? Is it relevant to this study?

2.  Please remove the redundance in Lines87-88.

3.  At the end of the Introduction section, a short description about the structure of the document is recommended to inform the reader about how it is organized and what to expect in the next sections.

4.  In Lines 138-143, I think more information on the images (resolution, color depth, ...) is required, as well as some details on the lighting conditions of their capture.

5.  In Lines 144-147, I think the use of 'augmentation' regarding the dataset should be explained in more detail: are synthetic images being used?

6.  The choice for the architectures (and their versions) used across the manuscript should be explained and justified in detail regarding the specific intended application.

7.  The versions and other relevant details of all the software packages used should be provided in the document.

8.  Please, justify the choice for the hyperparameters mentioned in Lines 186-191.

9.  In Lines 210-214, how was the choice made for the algorithms for separating the lesion pixeles from the undamaged skin?

10.  Please, revise Figure 5 for a better readability of the texts and images, possibly increasing the size of both. Additionally, this figure seems not to be very informative and perhaps should be enriched with information about the images being processed. Check also Figure 6.

11. No architectural diagrams are provided for the different algorithms/networks used in the work, with only text descriptions of them. Please consider including them, at least for those configurations that resulted to be more convenient for the goals of the manuscript.

12.  From a practical point of view, what could be the cost and work needed for implementing a system like this in a real farm? How would be the data recovery from each animal and the device battery charge made? What would be the investment needed?

13. I consider that the authors should include a proper Discussion section, which should clearly determine what is the main contribution of the article compared to other studies or similar works in the related field of study. Authors need to pay special attention to this comparison and highlight the relevance of their contributions. Also, a critical analysis of their proposal should be included there.

Round 2

Reviewer 2 Report

Comments and Suggestions for Authors

The Authors have provided a new version of the manuscript with some changes from its previous version, which aim to address the issues pointed to by the Reviewers.

The Authors have provided as well a letter containing the answers to the Reviewers' comments. This letter is intended to provide appropriate answers to the questions posed by the Reviewer, detailing which changes were made in the document as an answer to each question, and their expected contribution to the quality of the work. However, I think the Authors failed to provide proper answers to several of my concerns, lacking mentions to the specific changes made to the document. Additionally, line numbers in the cover letter do not match the ones in the revised manuscript document. All that somehow difficulted the review work

Please find next the main concerns that I have regarding the answers to my previous comments to the initial manuscript:

Comment 1: In Lines 19-21 the Authors state: 'A time-of-19 flight (Microsoft Kinect V2) camera captured the top-down depth and RGB images of sows in farrowing crates.' I could find no other mention in the document about such top-down depth captured. Has it been used? Is it relevant to this study?
Response 1: No, the depth information from the sensor has not been used. Both the RGB and the depth images were collected as this data collection was done for a larger study but only the RGB images were used for this experiment.

Answer: I still think that the mention to 'time-of-flight' should be removed, as it is not relevant to the work and might be misleading.

Comment 3. At the end of the Introduction section, a short description about the structure of the document is recommended to inform the reader about how it is organized and what to expect in the next sections.
Response 3: The section enlisting the structure has been added on lines 116-120.

Answer: Line numbering is not the same for the capture in the cover letter as in the revised document. Section 1 should also be mentioned in that structure. I do not think the response provided addresses the issue pointed to in my comment. Blank Line 162 is unnecessary. Please check Line 167 for consistency.

Comment 4. In Lines 138-143, I think more information on the images (resolution, color depth, ...) is required, as well as some details on the lighting conditions of their capture.
Response 4: The required changes have been made to the manuscript with the details of lighting on lines 133-134. Image details on 140-142.

Answer: I could not find specific information on the lighting in the document.

Comment 5. In Lines 144-147, I think the use of 'augmentation' regarding the dataset should be explained in more detail: are synthetic images being used?
Response 5: No, no synthetic images were added to the dataset. Images from the first week of farrowing did not show any signs of lesions and were retained to detect false positives more accurately. The term ‘augmentation’ has been replaced with ‘increased’ which might help with confusion.

Answer: I would recommend using 'extended' instead of 'included' for that replacement. In addition, a new mention to 'augmenting' is present in Line 471.

Comment 12. From a practical point of view, what could be the cost and work needed for implementing a system like this on a real farm? How would be the data recovery from each animal and the device battery charge made? What would be the investment needed?
Response 12:  

Answer: It seems this comment has been left unanswered.

Additionally, it seems that the '5. Conclusions' title is wrongly formatted. Also, now that Conclusions section seems too short for its intended goal:  providing a honest, impartial and accurate criticism of the achievements claimed from the article is expected, including the specific dimension and the limitations of the results obtained, as well as the future lines of work open as a consequence of the efforts made.
